# Use of Asset Administration Shell Coupled with ISO 15926 to Facilitate the Exchange of Equipment Condition and Health Status Data of a Process Plant

Bongcheol Kim [1], Seyun Kim [1], Hans Teijgeler [2], Jaehyeon Lee [3], Ju Yeon Lee [4], Dongyun Lim [5], Hyo-Won Suh [6] and Duhwan Mun [1],*

1   School of Mechanical Engineering, Korea University, 145 Anam-ro, Seongbuk-gu, Seoul 02841, Korea
2   OntoConsult, Laanweg 28, 1871 BJ Schoorl, The Netherlands
3   Smart Manufacturing Innovation Center, Korea Electronics Technology Institute, 42, Changeop-ro, Sujeong-gu, Seongnam-si 13449, Gyeonggi-do, Korea
4   Department of Mechanical System Design Engineering, Seoul National University of Science and Technology, 232 Gongneung-ro, Seoul 01811, Korea
5   R&D Center, Safetia Co., Ltd., 201, Prugio-City 1st, 2 Jeongja-ro, Bundang-gu, Seongnam-si 13560, Korea
6   Department of Industrial & Systems Engineering, Korea Advanced Institute of Science and Technology, 291, Daehak-ro, Yuseong-gu, Daejeon 34141, Korea
*   Correspondence: dhmun@korea.ac.kr; Tel.: +82-2-3290-3359

**Abstract:** The digital twin has emerged as a crucial technology for smart production. The Asset Administration Shell (AAS) is a standard tool that can support the digital representation of a process plant. We present a method to use AAS coupled with ISO 15926 to facilitate the exchange of maintenance data in process plants. To accomplish this, the operation and maintenance (O&M) system framework employed in process plants is defined. Information requirements are derived based on this framework, and a maintenance data structure is designed. Along with this, reference data are applied to identify the types of equipment and properties of each equipment type using ISO 15926. According to the pre-designed data structure, a neutral format based on AAS is developed to exchange maintenance data among software systems in O&M. The neutral format is verified through a test case of exchanging maintenance data (equipment condition and health status data) in terms of applicability to O&M systems in the field and compliance with the AAS meta-data model.

**Keywords:** Asset Administration Shell; CMMS; data exchange; ISO 15926; reference data; smart production

## 1. Introduction

Industry 4.0 is a new manufacturing paradigm that can potentially enhance technology, industry, social patterns, and processes in the 21st century due to increased interconnectivity and smart automation [1]. In Industry 4.0, a digital twin, which is a synchronized virtual object corresponding to a physical asset, is required. When the latest information and communication technologies, such as the Internet of Things, edge computing, artificial intelligence, big data, and cloud, are integrated with a digital twin, real-time awareness, granular control, and intelligent decision-making become possible through a two-way information flow between equipment, products, and systems.

Reference Architecture Model Industry 4.0 (RAMI 4.0) is the reference architecture for the implementation of Industry 4.0. RAMI 4.0 is a three-dimensional map showing how to address the issues of Industry 4.0 in a structured manner and provides a common background to promote the mutual understanding of all stakeholders related to Industry 4.0. In RAMI 4.0, a management shell must be built to manage physical assets. The Asset Administration Shell (AAS) constitutes a digital representation of a physical asset and is deemed essential to the implementation of a digital twin for Industry 4.0. It performs the

roles of asset, communication, information, and function layers in RAMI 4.0. It further provides a meta-data model that has an information structure supporting data exchange between assets and data gathering of physical assets, such as equipment, parts, materials, software, and documents [2].

The concept of AAS was first proposed by Zentralverband Elektrotechnik- und Elektronikindustrie eV (ZVEI) in 2016 [3]. Platform Industrie 4.0, which is a digital manufacturing network in Germany, released a technical document on the meta-model of AAS in 2018 along with ZVEI [4]. The latest version of this technical document is 3.0 [5]. Wei et al. proposed a direction for the research on AAS from three perspectives [6]: Interoperability, modeling, and industry application. Son et al. analyzed and classified the studies on the application of a digital twin per product lifecycle stage defined in RAMI 4.0 [7]. Wu et al. proposed a service-oriented architecture for the exchange of cloud-based design and manufacturing data [8]. Cavalieri and Salafia proposed a predictive maintenance model combined with AAS for optimizing the availability of equipment and devices while securing the interoperability of production data in the manufacturing industry [9]. Lang et al. studied how to use the AAS as a tool to assist workers in performing maintenance tasks. In addition, eClass and IEC 61360 were suggested as reference data that can be connected to AAS [10]. Yan and Duan analyzed the correlation between product quality characteristics and real-time fluctuation of equipment operating conditions in a smart manufacturing environment and proposed a product-quality prediction model [11]. Toro et al. proposed an implementation method of an AAS-based mini factory model for robot modules and introduced the implemented cases of AAS models for robot workstations [12]. William Motsch et al. proposed a model of AAS-based cyber-physical production modules for electrical energy consumption interfaces and implemented a prototype system [13]. Furthermore, Ye et al. proposed an AAS-based data exchange method for enterprise resource planning (ERP) and a manufacturing execution system used in the manufacturing industry [14]. Rauh et al. considered artificial intelligence (AI) models used in the manufacturing field as assets and studied how to manage the meta-data of each AI model using AAS [15]. OpenAAS was developed by the RWTH Aachen University as the development environment for an industrial control system that supports AAS [16]. The open-platform communications unified architecture (OPC UA) has been utilized as a data communication method for AAS data among various systems, including the cloud on a network [17]. Pribiš et al. suggested a method to use the AAS by connecting it to OPC UA architecture server [18].

Process plant projects involve a variety of stakeholders including plant owners, EPC contractors, and equipment suppliers. Different systems are used depending on engineering disciplines and lifecycle phases. Each system has its own internal data structure. In this environment, various problems such as data omission, data inconsistency, and data redundancy occur when exchanging data between different engineering systems. To solve this problem, a method of defining a neutral data model independent of application systems and using it to exchange data between various engineering systems is generally adopted in the industry.

The cost of data interoperability issues is highest in the O&M phase [19]. Factors that make data exchange in the O&M phase difficult include the use of various equipment manufacturing by different manufacturers even for the same purpose, frequent replacement of equipment and components in the O&M phase, and the upgrade or change of maintenance systems during the long plant running lifecycle.

Klose et al. studied a method to operate process plants efficiently by ensuring the flexibility and reusability of the modularized process plant data with the process orchestration layer [20]. Wiedau et al. studied the application of industrial data standards including AAS, AutomationML, Capital Facilities Information Handover Specification, Data Exchange in the Process Industry, and ISO 15926 to AI technologies so that the process to integrate the lifecycle data of a process plant can be enhanced [21]. Akinyemi et al. suggested a data integration framework in a process plant based on ISO 15926 to determine the reusability of decommissioned items and integrate heterogeneous data in the oil and gas industry [22].

Igamberdiev et al. suggested multi-level modeling to integrate complex heterogeneous data in the oil and gas industry [23]. The formalization, implementation, querying, and verification of multi-level modeling based on ISO 15926 and OSA-EAI of MIMOSA were studied to secure data interoperability. Mun et al. developed a neutral data model by extending the generic product model developed by Hitachi, Ltd. to transfer design data to an O&M phase in a nuclear power plant [24]. Based on the data model, a neutral data warehouse was constructed to bridge EPC and O&M phases in the context of engineering data.

Most studies related to AAS have targeted hardware utilized in industries that have adopted a discrete production method, such as general manufacturing, robot, and electrical and electronic industries. In contrast, this study proposes a data exchange method based on AAS between software systems that support maintenance for process plants that have a continuous production method. The system framework that supports the operation and maintenance (O&M) of a process plant is defined in order to identify software systems related to maintenance. Furthermore, an AAS meta-data model is used to design a maintenance data model for a process plant, and a method for exchanging data using such a model is developed. When a maintenance data model is defined, clear and consistent identification of the elements of the submodels constituting the model is required. Accordingly, a method is developed for extending the reference data provided in the ISO 15926 standard and using them with the elements of the submodels.

The scope and procedure of AAS-based maintenance data exchange coupled with ISO 15926 are shown in Figure 1. First, an AAS-based neutral equipment maintenance data model is defined. The authors analyze data used in various maintenance systems and select the type of data to be applied in this study. Each data item selected is classified as an asset, property group, or property. If an item is an asset, an asset is added. If the asset has an assembly relationship with other assets, a BOM submodel is added. If an item is a property group, a submodel is added. If an item is a property, a submodel element is added. For detailed specification of the submodel element, a concept description is added to connect corresponding external reference data to the element. Once the AAS-based neutral equipment maintenance data model coupled with ISO 15926 is defined, the grammatical integrity is checked using the AASX package explorer. After the completion of model verification, an experiment of maintenance data exchange is performed using the neutral data model. In the experiment, after exporting data created in one maintenance system in the AAS format, the data are imported to another maintenance system followed by examining whether the data are exchanged correctly.

The contribution of this study is as follows. First, the AAS was applied to the exchange of equipment condition and health status data between process plant maintenance systems beyond the existing hardware-centered applications of the AAS. Second, a neutral maintenance data model for process plants was defined based on the AAS meta-data model. In this process, a method to connect the ISO 15926 reference data library to the neutral model was presented to describe equipment classifications, equipment properties, and units in detail. Third, to identify maintenance data to be exchanged, systems used for O&M in a process plant were apprehended, data generated in each system were identified, and the flow of the data was analyzed. To the best of our knowledge, few studies have been conducted on the exchange of maintenance data of a process plant using an AAS meta-data model. Therefore, this study makes a significant contribution by proposing a method for exchanging maintenance data using AAS coupled with ISO 15926 RDL.

The remainder of this paper is organized as follows. In Section 2, the O&M framework of a process plant is explained. In Section 3, a method for exchanging maintenance data in this framework based on AAS is proposed. In Section 4, extension and connection methods are proposed for ISO 15926-based reference data. In Section 5, example cases of exchanging maintenance data using a maintenance AAS model are discussed. Lastly, Section 6 concludes the paper.

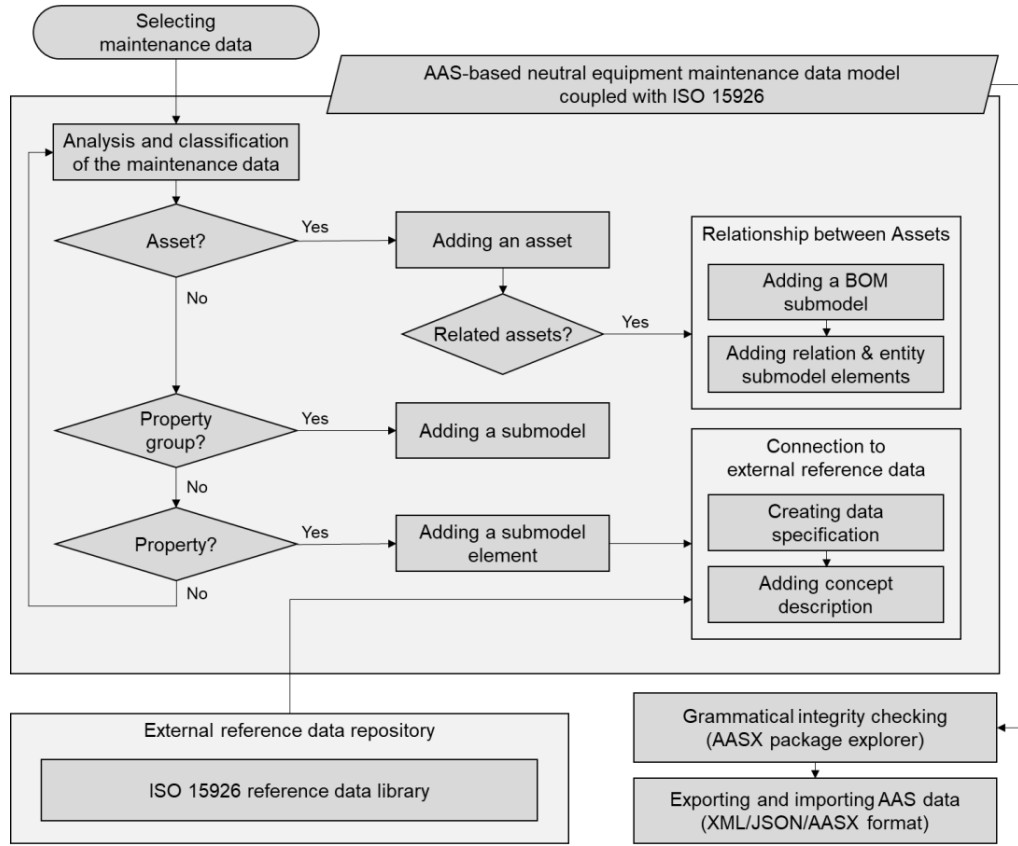

**Figure 1.** Research scope and procedures.

## 2. O&M System Framework

Various O&M systems have been utilized for the management of process plants. Efficient and optimized running requires the generation of accurate O&M data, integration and exchange of O&M data, as well as secured interoperability between data. To explore where maintenance data are generated and what systems such data are shared with from the perspective of the running of a process plant, the configuration of various O&M systems applied to a process plant was analyzed, as shown in Figure 2. O&M systems of a process plant largely consist of the *Plant O&M Data Warehouse* and four system groups. The *Plant O&M Data Warehouse* collects, connects, and integrates all data generated during the lifecycle of a process plant to provide the information required in other systems in a timely manner. The *Process Plant* group generates O&M data obtained from all sensors and equipment installed on site and includes maintenance works performed on site. The *Resource and Production Management Systems* group supplies raw materials for product production, comprehensively manages enterprise resources including equipment, manpower, and materials, and establishes and manages production plans of products. The *Operation Systems* group monitors the production process status and controls processes by simulating the optimal output condition. Lastly, the *Maintenance Systems* group analyzes the state of assets such as equipment, performs necessary maintenance required for the safe use of such assets, and manages maintenance history.

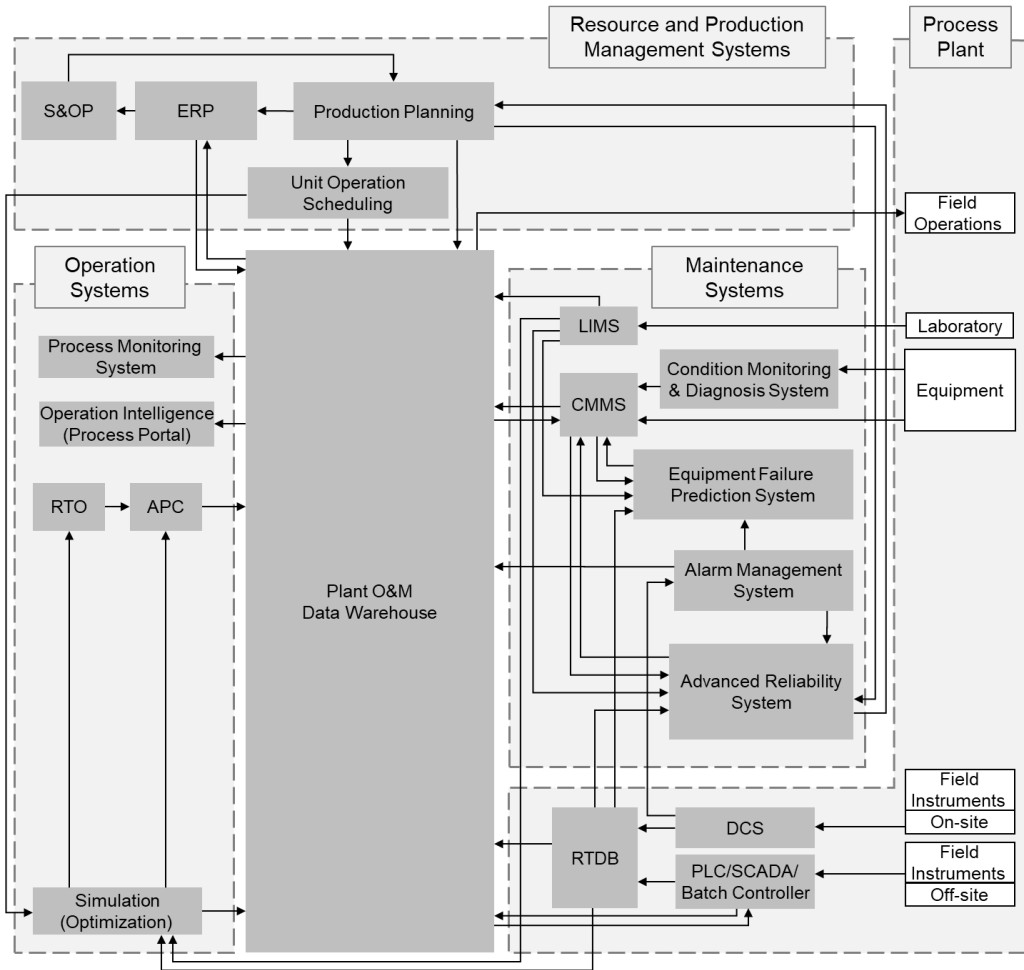

**Figure 2.** Configuration of O&M systems in process plants.

As shown in Figure 3, various maintenance systems connected with respect to the *Computerized Maintenance Management System (CMMS)* are used in the maintenance of a process plant. The *Laboratory Information Management System (LIMS)* tracks and manages the properties of materials and intermediate outputs that are periodically sampled in a process plant. The result of property analysis is used for lifecycle prediction or process optimization. The *Alarm Management System* collects all alarms generated in the field, filters the information according to pre-determined criteria, and designates a hazard class. The *Condition Monitoring & Diagnosis System* collects and analyzes the equipment condition data through sensors attached to major equipment constituting a process plant. The *Equipment Failure Prediction System* predicts the failure point of equipment based on the status, alarms, and maintenance history of the equipment, and establishes a preventive maintenance plan. The *Advanced Reliability System* predicts future changes in equipment performance according to the changes in processes based on the data warehouse in which O&M data are integrated. *CMMS*, which plays a key role in the maintenance of a process plant, receives and manages equipment status, alarms, failure predictions, and optimized maintenance work orders from different maintenance support systems; *CMMS* also manages the equipment maintenance history and parts inventory and controls maintenance work performed by field workers.

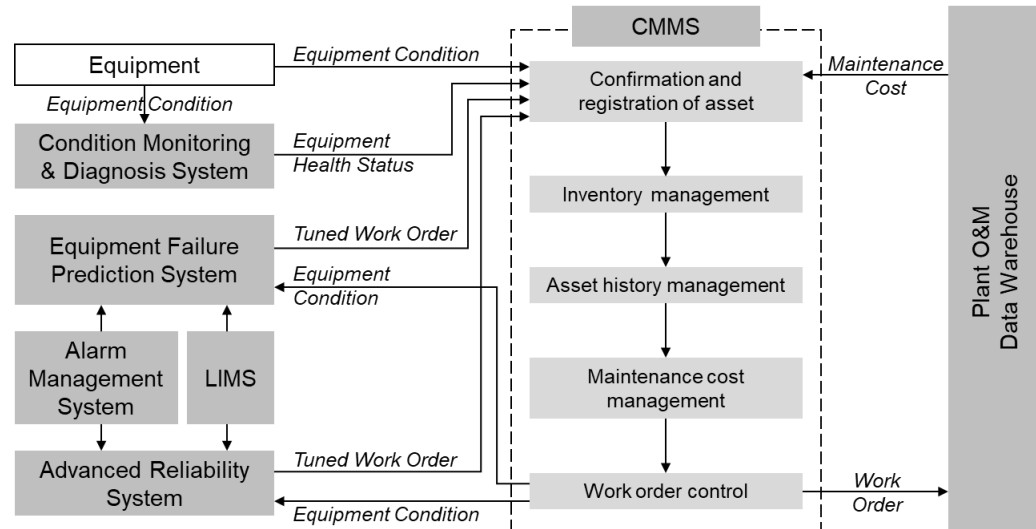

**Figure 3.** Maintenance systems and data exchanged between them.

## 3. AAS-Based Exchange of Maintenance Data

Maintenance systems used in the running process of a process plant contain various types of data that are frequently exchanged between systems. An efficient data exchange method is required given that large amounts of time and costs are involved in the data exchange process. Indirect exchange using a neutral data model is an effective data exchange method for multiple systems [25–27]. In this section, a method for establishing a neutral model using an AAS meta-data model is explained, and a neutral data model developed for exchanging maintenance data of a process plant is proposed.

### 3.1. AAS Meta-Data Model

The AAS meta-data model consists of five parts, as shown in Figure 4. The AAS part comprises *AssetAdministrationShell* and *Security* [5]. *Security* describes the matters related to data security. *AssetAdministrationShell* is a top-level object representing a digital twin corresponding to a physical asset, and has a unique ID. The Asset part includes data and features related to assets. *Asset* refers to identifiable assets. *IdentifierKeyValuePair* expresses a valid ID of the asset. In addition, *Asset Kind* indicates the type of asset. Asset object, asset ID, asset type, and detailed information about the asset are connected through *AssetInformation*. The Submodel part includes the information per domain or technology sector of the asset. *Submodel* represents the group of properties belonging to the same domain or same technology sector of the asset. *Submodel* includes the description and type of a property group. Given that one asset has various property groups, *AssetAdministrationShell* consists of multiple *Submodels*. The properties of *Submodel* are expressed as *SubmodelElement*, in which one *Submodel* has multiple *SubmodelElements*. The Property part includes the types of various properties that may belong to the asset. *Entity* is the lower-level element of *SubmodelElement* used for expressing an entity. Entity refers to the generated *Asset*, *Submodel*, and *Property*, and includes the identifier and type information of an entity. *DataElement* is the lower-level element of *SubmodelElement* representing property values and incorporates *Property* and *RelationshipElement*. *Property* is used to express general property values and includes *Value* and *Value Type*. *RelationshipElement* is used to express the relationship between two elements. The External Reference part manages the external reference data list. *ConceptDescription* is used to identify external reference data or internally defined reference data. *ConceptDescription* solely identifies the type of *Submodel* or *Property* or related units independently and permanently to a system. The list of reference data used in *AssetAdministrationShell* is managed in the form of a *ConceptDescription* list. Detailed information on external reference data is stored in *DataSpecification*. Specifically, *DataSpecification* includes

the information related to reference data defined in other standards such as IEC 61360 CDD (common data dictionary), ISO 15926 RDL (reference data library), and eClass.

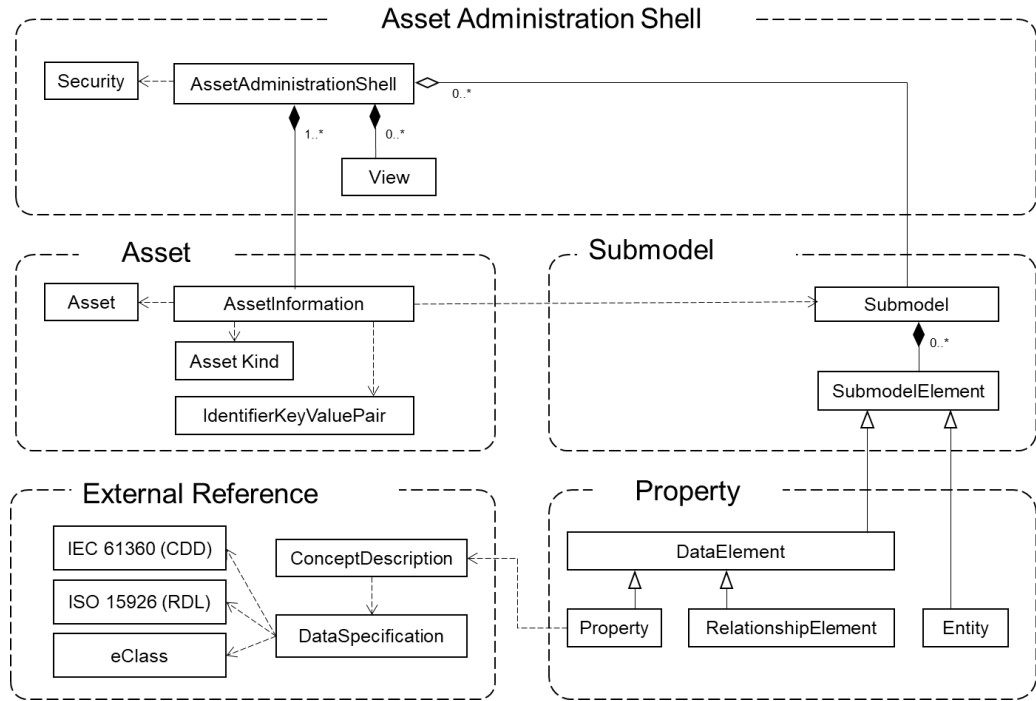

**Figure 4.** Meta-data model of AAS.

The structure of the AAS data model defined using the meta-data model explained above is shown in Figure 5. *AAS* consists of a Header and a Body. The Header defines the identifier of *AAS* and *Asset*. The Body defines *Submodels* of *Asset* and *SubmodelElements* such as *Property*. The defined *Submodels* are managed in *ComponentManager*. The semantics of *Submodel* or *Property* are determined using external reference data connected through *ConceptDescription*. The correlation between AASs is formed at the *Asset* or *Submodel* level [28].

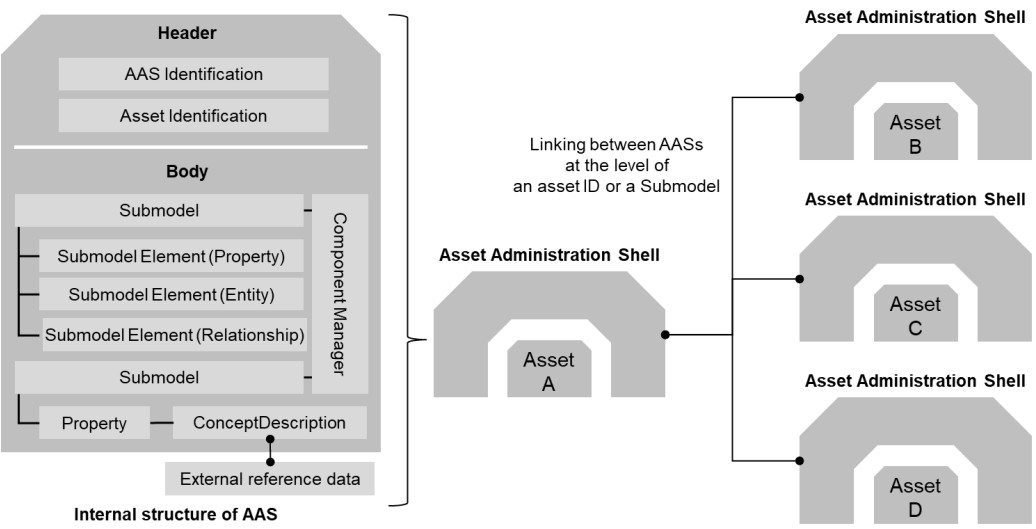

**Figure 5.** Model definition using the AAS meta-data model.

### 3.2. AAS-Based Maintenance Data Exchange between Process-Plant O&M Systems

The process of exchanging maintenance data based on AAS between different process plant O&M systems is shown in Figure 6. When data are transferred from System A to System B, *System A* extracts the data being delivered from a native database. Then, the data interface of *System A* configures the AAS data package. Here, the package refers to a set of all information items (*Asset*, *Submodel*, etc.) composing AAS, whose data are encoded according to methods specified in technical documents in the form of extensible markup language (XML), JavaScript object notation, or a resource description framework. Once the AAS data package is prepared, the data server performs data security processing and then sends the data package to *System B*. The data client of *System B* receives the data package sent by the data server of system A and then performs verification. Subsequently, the data interface of *System B* converts the AAS data included in the package into a native data format. Lastly, the converted data are merged or updated in a native database of *System B*.

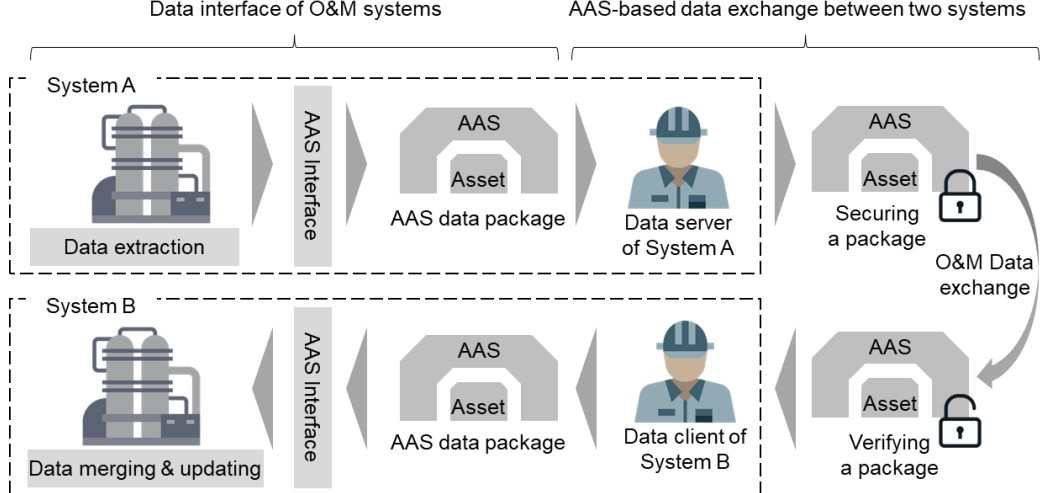

**Figure 6.** AAS-based data exchange between different O&M systems.

### 3.3. AAS-Based Neutral Process-Plant Maintenance Data Model

Maintenance data shared between O&M systems include the equipment condition, equipment health status, work orders, tuned work orders, and maintenance cost. Additionally, the maintenance history is managed within CMMS. This study specifically defines an AAS-based model for expressing maintenance data related to equipment condition and equipment health status. The equipment condition and equipment health status contain information on the operation status, equipment property, damaged condition, and hazard class. These parameters are affected by the process unit on which the equipment is installed, the equipment itself, and the major components of the equipment. Therefore, AAS for the process unit, equipment, and component must be declared when defining an AAS-based model, and then submodels should be designed for each AAS. Finally, the assembly relationship between the process unit, equipment, and component is established.

Considering the points addressed above, the result of defining an AAS-based neutral model of process plant maintenance data is shown in Figure 7. Here, the neutral model consists of three AAS types of process unit, equipment, and component. It is generally recommended that an AAS has four main submodels of Identification, Technical data, Operational data, and Documentation. Identification includes the identification of an asset and manufacturer (or supplier) information. Technical data have the technical specifications of an asset. Operational data include real-time data collected from the field. Documentation includes various types of documents such as drawings, specifications, and manuals. *AASs* of a Process Unit, Equipment, and Component have the Identification *submodel* to describe the identification of each asset. Technical data *Submodel* of Process Unit *AAS* includes the

average cost of equipment included in the process, production loss cost occurring during maintenance, and management costs of human and environmental resources, process flow diagram (PFD) ID, and piping and instrument diagram (P&ID) ID. Documentation *Submodel* of Process Unit *AAS* contains documents such as the PFD and P&ID drawings. Technical data *Submodel* of Equipment *AAS* includes the equipment type and installation date. Furthermore, it also includes the property information of the materials used in the equipment, design temperature/pressure, operating temperature/pressure, and test history. Technical data *Submodel* of Component *AAS* includes the ID of equipment containing a corresponding component, component type, size, design lifecycle, installation date, and operating environments such as the allowable temperature and allowable pressure. Furthermore, it also includes information on the degree of equipment damage in terms of parameters such as corrosion and fatigue or hazard. Operational data *Submodel* of component *AAS* includes real-time data such as temperature, pressure, and flow supply measured during the operation. The assembly relationship between the process unit, equipment, and component is expressed as the BOM Aggregate, which is one of the *Submodels.* The type of BOM Aggregate is *billOfMaterialRef*. This *Submodel* consists of *Entity*, which indicates targets to be assembled, and *Relation*, which indicates assembly conditions.

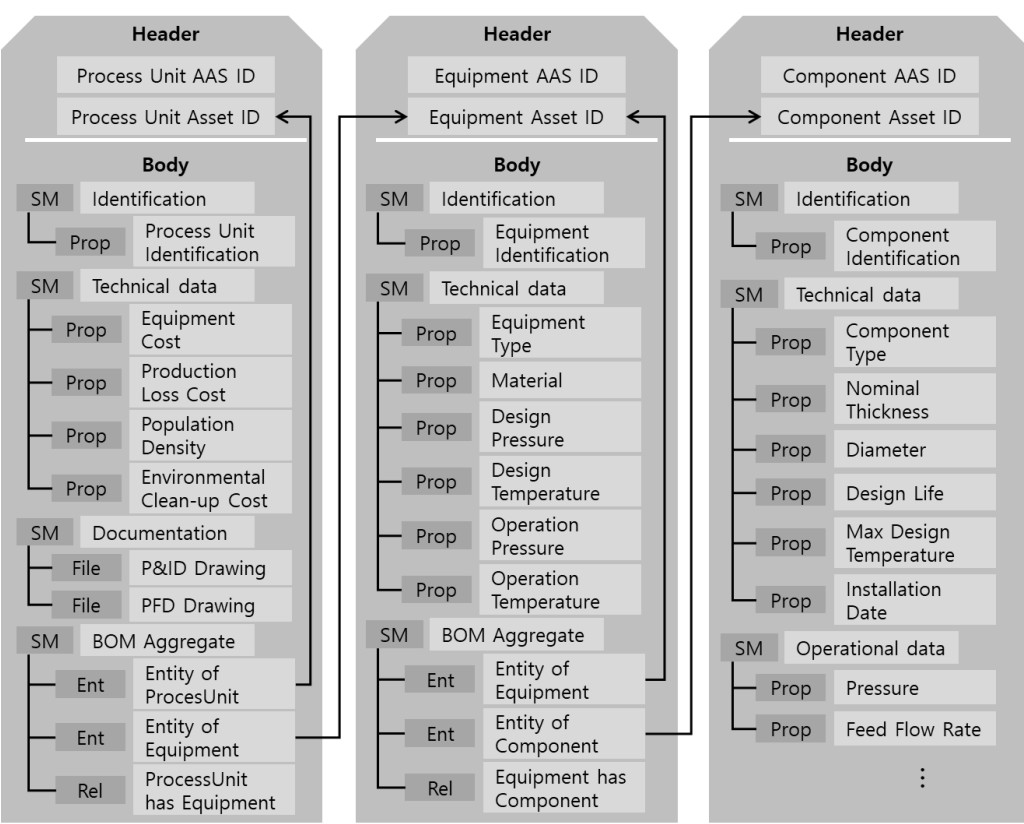

**Figure 7.** AAS-based neutral model for process plant's maintenance data.

## 4. Extension of ISO 15926 Reference Data to Be Used with AAS

Reference data can be considered as a common dictionary that defines classes, properties, units, and activities that are frequently used in a process plant. The reference data are employed to assign a unique and consistent identifier for the classification of equipment, properties, and units that are generally used in application systems. Well-known reference data include ISO 15926 RDL, IEC 61360 CDD, and eClass. ISO 15926 provides information resources and implementation methods for exchanging, sharing, and integrating life cycle data of a process plant [24]. Major parts of ISO 15926 include a conceptual data model in part 2 [29], reference data of topology and geometry in part 3 [30], reference data in part 4 [31], a template method in part 7 [32], and a web-ontology-language implementation in

part 8 [33]. IEC 61360 CDD provides a data dictionary and formalized descriptions for electrical and electronic products and services [34,35]. eClass is managed by the Industry Consortium ECLASS e.V. Association and provides reference data and search services for classifying industry products, materials, and services [36].

AAS essentially recommends using eClass among the reference data explained above. Furthermore, AAS internally supports IEC 61360 CDD. However, the reference data provided by eClass or IEC 61360 do not fully cover the piping, mechanical, structure, HVAC, electric, and instrumentation disciplines that constitute a process plant. By contrast, ISO 15926 RDL provides an ample amount of reference data specialized for a process plant. Therefore, in this study, we applied ISO 15926 RDL as external reference data.

ISO 15926 specifies the method for constructing an RDL that expresses common information objects (class, property, unit, activity, etc.) of a process plant. It also provides the initial reference data positioned at the very top of an RDL. When defining reference data in ISO 15926 RDL, reference data ID, type, label, definition, classification (superclass/subclass), and membership (classifier/classified) must be defined. When searching for the reference data saved in ISO 15926 RDL, search tools such as the 15926 browser [37] or a reference data service [38] of POSC Caesar are used, as shown in Figure 8.

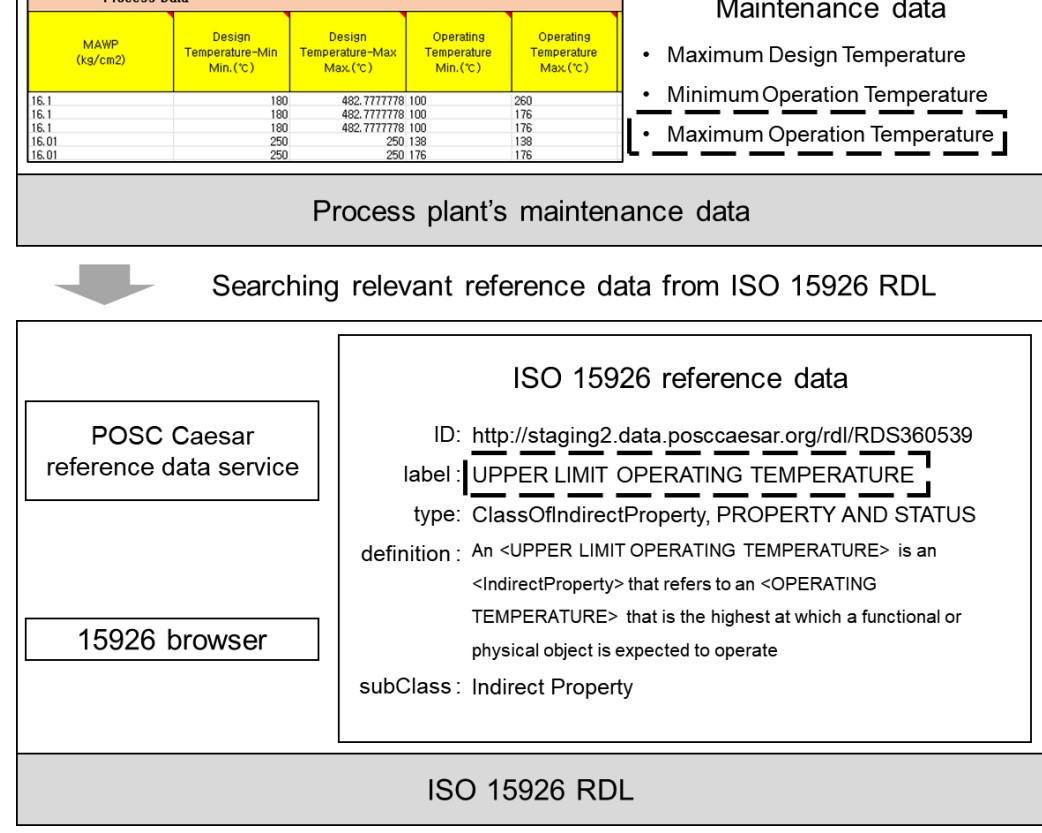

**Figure 8.** Searching ISO 15926 reference data through web.

When defining an AAS-based neutral process plant maintenance data model, there were a high number of cases in which reference data corresponding to equipment types, properties, or units of measure were not present in ISO 15926 RDL. Accordingly, ISO 15926 RDL was extended to cover situations in which it was difficult to clearly describe maintenance data with existing reference data. As shown in Figure 9, the RDL can be extended through the definition of new reference data by inheriting existing ones. A total of 39 new reference data items were added when defining the AAS-based neutral process

plant maintenance data model. Any such extension can be made by setting up a local RDL extension. For example, the code for adding Arid Atmosphere is shown in Listing 1.

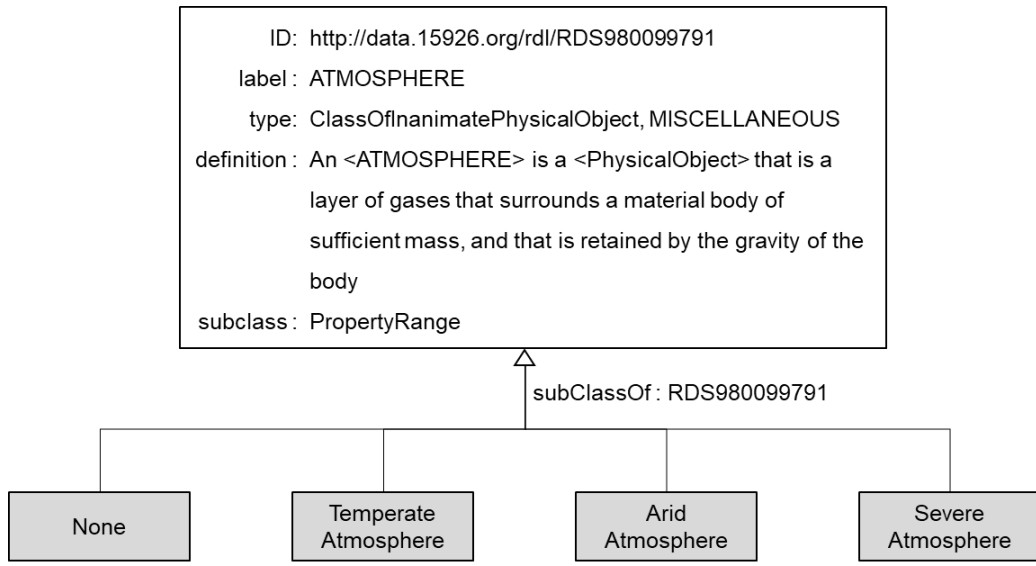

**Figure 9.** Extension of ISO 15926 reference data.

**Listing 1.** Code to define Arid Atmosphere in a local RDL.

| | |
|---|---|
| id | http://rdl.example.org/R438341 |
| rdfs:label | ARID ATMOSPHERE |
| rdf:type | ClassOfInanimatePhysicalObject |
| rdfs:subClassOf | ATMOSPHERE |
| skos:definition | An <ARID ATMOSPHERE> is an <ATMOSPHERE> that is characterized by an overall moisture deficit, often expressed as annual precipitation being less, often significantly less, than potential evapotranspiration. (source: Oxford Bibliographies) |
| meta:valEffectiveDate | 2021-10-28T00:00:00Z |

The method for connecting ISO 15926 reference data in the AAS-based neutral process plant maintenance data model is shown in Figure 10. *ConceptDescription* is used to express the semantics of a property declared as a *Property*. To accomplish this, *ConceptDescription* has an ID for connecting external reference data. Another *ConceptDescription* may be referred to by the corresponding *ConceptDescription* to represent the semantics in detail. For example, when defining a property with a unit of measure, *ConceptDescription* of a property refers to unit-of-measure *ConceptDescription*, as shown in Figure 10.

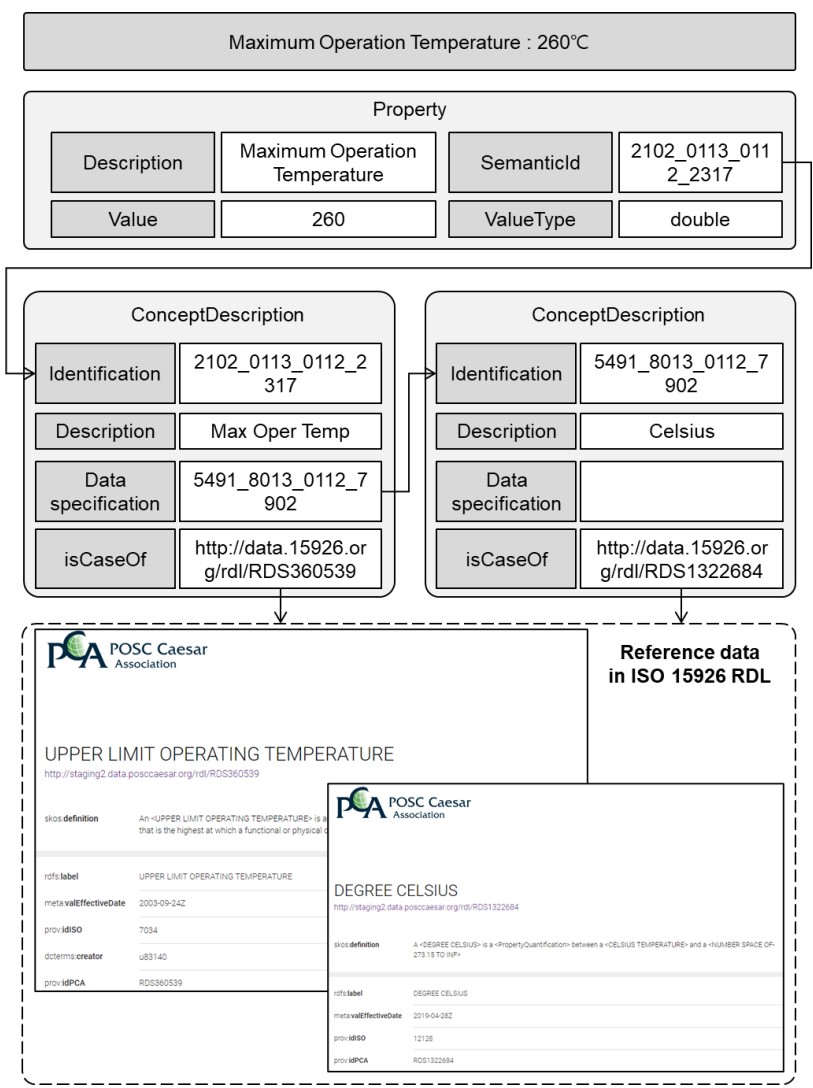

**Figure 10.** Connection between ISO 15926 reference data and an AAS-based neutral model.

## 5. Case Study: Exchange of Equipment Condition and Health Status Data

### 5.1. Data Preparation and AAS Configuration

For the proof of concept of the proposed AAS-based data exchange method, maintenance data provided by the 'H' oil company were utilized. For the experiment, equipment condition data and health status data of the process equipment diagnosis system (PEDS) were sent to a 3D-based portal system for plant O&M using AAS-based neutral equipment maintenance data.

The data used in this experiment were equipment condition and health status data of three equipment elements constituting the residue desulfurization (RDS) process. The RDS process involves the desulfurization of sulfur, nitrogen, Conradson carbon residue, and metal from a high sulfur atmospheric residue (HS-AR). In the data exchange experiment performed, a reactor (MR-E18-05 55), tower (AS-E18-03), and vessel (MS-E18-14), which are the main equipment elements, were selected. Each of these elements has two reactors, three towers, and three vessels as sublevel components subject to maintenance. PEDS manages equipment condition and equipment health status data including equipment properties, operation status, damaged condition, and hazard class. With respect to the AAS-based neutral equipment maintenance model explained in Section 3, the result of configuring an AAS model for the RDS process, three equipment elements, and eight components is shown in Figure 11.

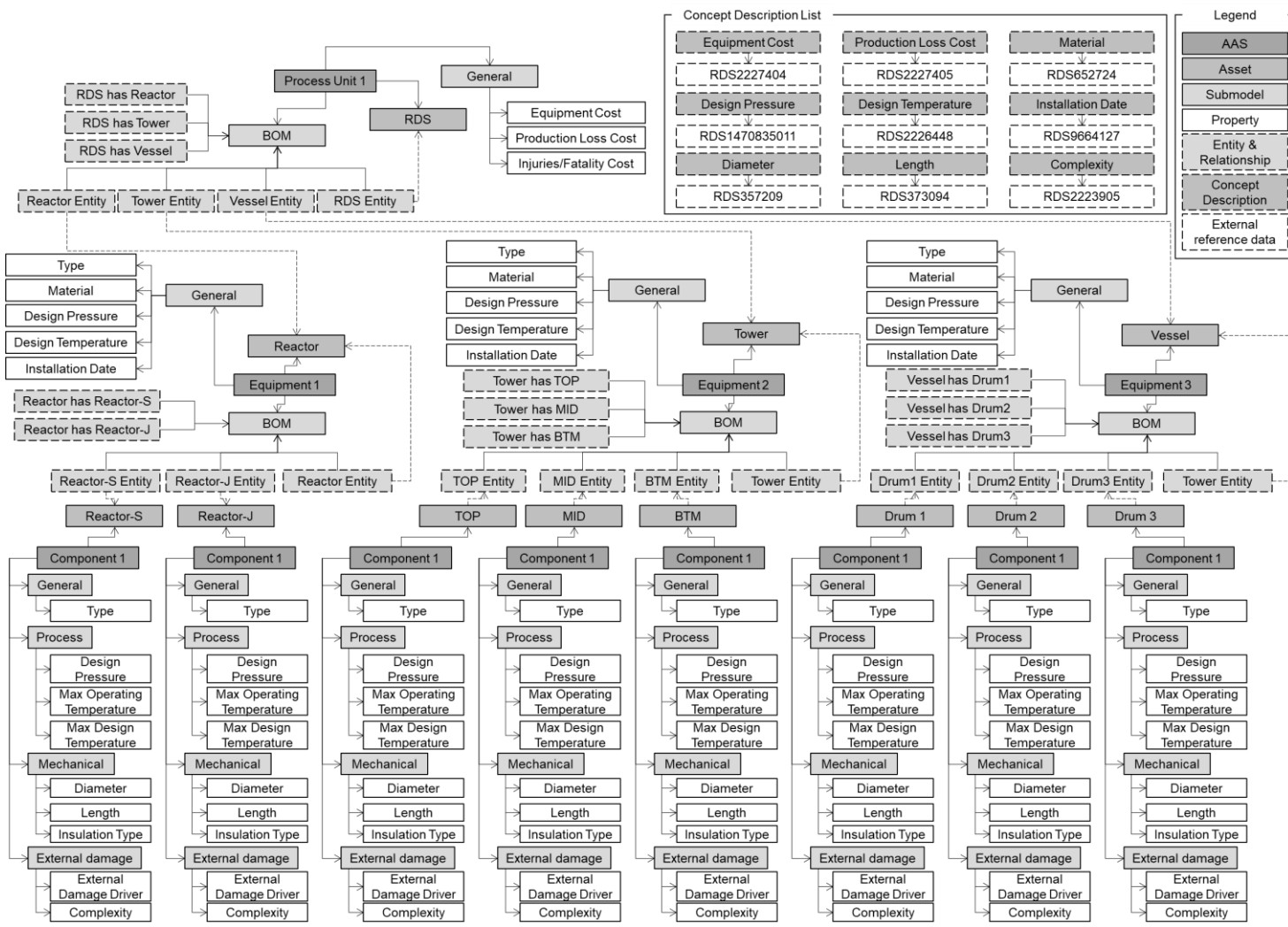

**Figure 11.** AAS configuration based on the neutral equipment maintenance data model for the RDS process used in the experiment.

### 5.2. AAS-Based Data Exchange Results

In the maintenance data exchange experiment, maintenance data extraction, AAS-based neutral data conversion, AAS data validation, AAS data transfer, and visualization were performed, as shown in Figure 12.

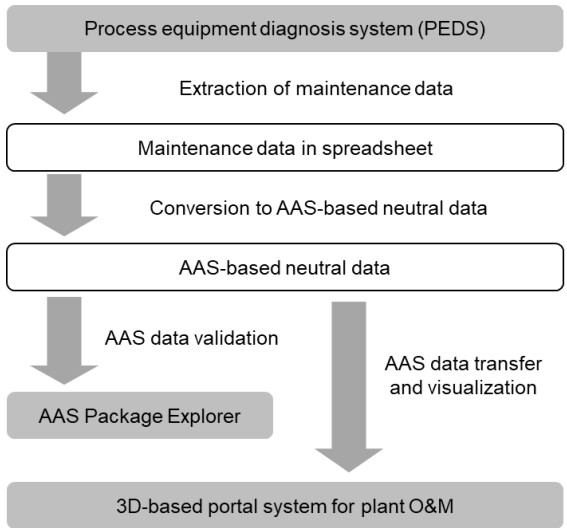

**Figure 12.** AAS-based data exchange procedure in the experiment.

After designing an AAS model for the RDS process, the equipment condition and health status data for such a process saved in the PEDS are converted to neutral model data in the XML format, as shown in Figure 13. After exporting the data saved in a relational database of the PEDS into a spreadsheet (Microsoft 365 Excel), a macro-based add-in module was implemented to convert AAS-based neutral equipment maintenance data.

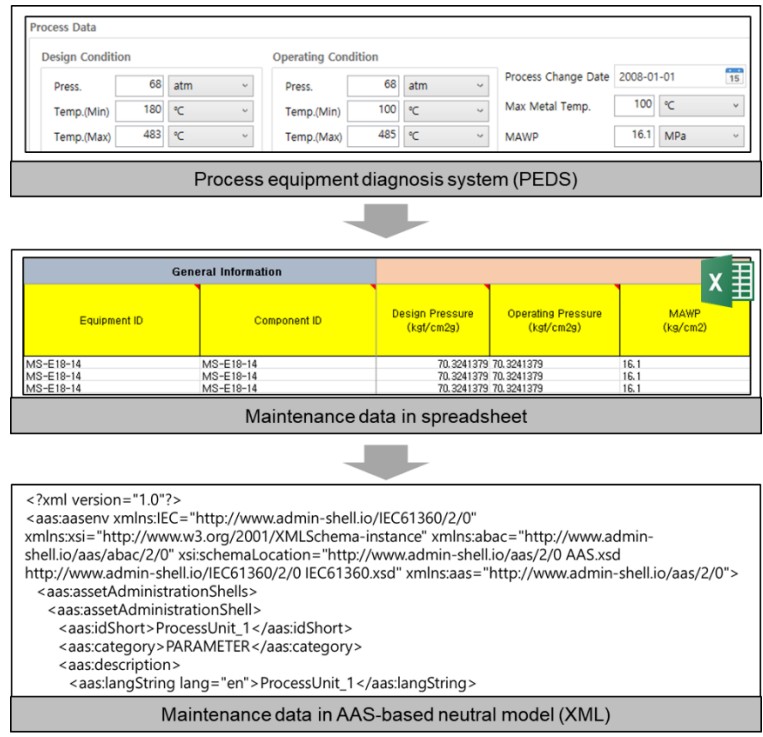

**Figure 13.** Maintenance data conversion to AAS-based neutral model data.

The grammatical integrity of the converted AAS-based neutral equipment maintenance data was confirmed by checking whether the neutral data could be loaded in the AASX package explorer [39], as shown in Figure 14. AAS Package Explorer is an open-source program developed in C# language for use as an AAS data editor. It runs on Windows 10 or later OSs. After loading the neutral data in the AASX package explorer, it was confirmed by comparing each item whether the data saved in the PEDS were accurately converted to the AAS-based neutral equipment maintenance data in terms of semantics. The result shows that IDs, assembly relationships, and properties of process/equipment/component matched between the native data of the PEDS and the neutral data.

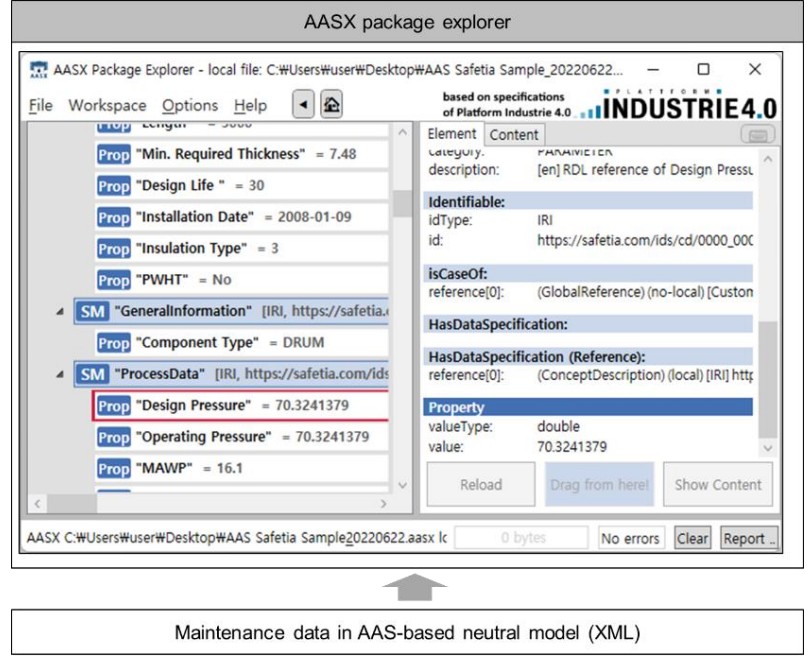

**Figure 14.** AAS conformance validation of converted neutral maintenance data.

After validation, the 3D-based portal system imported AAS-based neutral equipment maintenance data and visualized important items, as shown in Figure 15. The 3D-based portal system is a platform that provides O&M information along with related documents in an integrated manner on a 3D virtual plant model. Converted neutral maintenance data were linked to 3D objects of the virtual model, P&IDs, PFDs, and technical documents through equipment tag IDs. The experiment confirmed that neutral maintenance data exported from PEDS were correctly provided from the 3D-based portal system.

This study dealt with equipment condition and health status data that are part of maintenance data. However, according to the method presented in this study, the authors expect that the data model defined here can be extended to cover the entire maintenance data. AAS-based data exchange includes data model definition, connection to reference data, and the data exchange process. Among them, the connection to the reference and data exchange process is not closely related to the scope and type of exchange data. The data model definition using the AAS meta-data model consists of asset definition, submodel identification, submodel element specification, and connection between submodel element and reference data. This process of data model definition is not significantly affected by the scope and type of data. However, submodels, submodel elements, and reference data connected to the data model vary depending on the scope and type of data to be exchanged. In this matter, the data model defined in this study could serve as a cornerstone to develop an extended data model, which covers the entire maintenance data.

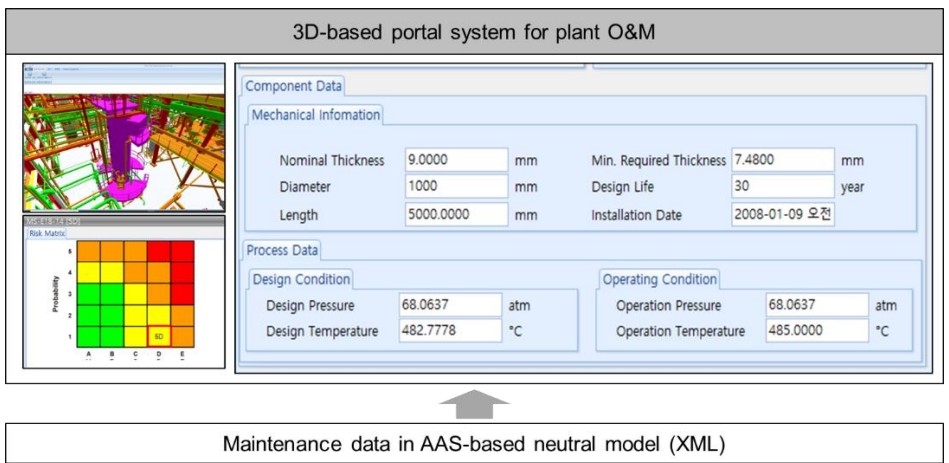

**Figure 15.** Maintenance data transfer to a 3D-based portal system.

## 6. Conclusions

Maintenance systems were identified, and maintenance data types were analyzed by defining the O&M system framework for a process plant. A neutral model specifying the maintenance data corresponding to equipment condition and health status was defined based on an AAS meta-data model. In this process, it was confirmed that defining equipment, specifying the hierarchical relationship between equipment, defining properties, and defining units of measure are possible with the AAS meta-data model. The reference data for identifying equipment types, properties, and units of measure were extended and built in ISO 15926 RDL to connect to the neutral model. Given that the scope of the reference data supported by each reference data standard varies, appropriate standards must be applied to each subject, and new reference data must be extended. In the data exchange experiment, the maintenance data for three types of equipment (a reactor, a tower, and a vessel) used in the RDS process provided by the 'H' oil company were saved in the PEDS and then converted to the AAS-based neutral equipment maintenance data. After generating the neutral data, it was read by the AASX package explorer to verify grammatical and semantic integrity. After validation, they were transferred and visualized on the 3D-based portal system. Accordingly, the possibility of applying the AAS-based neutral equipment maintenance data model to the O&M work of a process plant was verified. The exchange of maintenance data using the AAS-based data model enables data exchange independent from the equipment and maintenance system. Therefore, it is possible to transfer equipment data related to maintenance to a corresponding maintenance system in a neutral way and exchange the maintenance data between the maintenance systems regardless of equipment differences between manufacturers, the replacement of equipment, and upgrading or changing maintenance systems. In this study, an AAS-based data model was defined for equipment condition and health status data, and data exchange experiments were performed. In the future, the authors will conduct a study to extend this AAS-based data model to cover the entire maintenance data in a process plf.

**Author Contributions:** B.K.: Conceptualization, investigation, methodology, validation, writing—original draft; S.K.: Investigation, methodology, writing—original draft; H.T., D.L., and H.-W.S.: Resources, writing—review and editing; J.L. and J.Y.L.: Methodology, validation; D.M.: Conceptualization, methodology, writing—review and editing, supervision, project administration. All authors have read and agreed to the published version of the manuscript.

**Funding:** This research was supported by the AI-based gasoil plant O&M Core Technology Development Program (Project ID: 21ATOG-C161932-01) funded by the Korean government (MOLIT), the Liquid Air Energy Storage (LAES) and Utilization System Development Program (No. RS-2022-00143652) funded by the Korean government (MOLIT), and the Nuclear Power Plant Dismantling

Safety Reinforcement Convergence Core Technology Development Program (No. RS-2022-00155255) through the National Research Foundation of Korea (NRF) funded by the Korean government (MSIT).

**Conflicts of Interest:** The authors declare no conflict of interest.

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
