# Peer review of "Use of Asset Administration Shell Coupled with ISO 15926 to Facilitate the Exchange of Equipment Condition and Health Status Data of a Process Plant"

_processes, doi:10.3390/pr10102155_

Round 1

Reviewer 1 Report

The authors proposed the use of asset administration shell coupled with ISO 15926 to facilitate the exchange of maintenance data of a process plant.

However, the authors have not considered all types of maintenance data. In general, maintenance data is the data you share with the maintenance team so it can keep your assets up and running. It’s also the data you capture about your assets, including the work the team does on them, as well as the results of that work, in the form of maintenance metrics, key performance indicators, and reports. It seems to me that the authors only considered equipment condition and health status data, which greatly limits the applicability of the proposed method.

In addition, the literature review presented is quite poor. Most of the references cited in the work are not works published in periodicals, which compromises the verification of the novelty of the work.

Considering the above, I'm afraid I have to recommend the rejection of the work.

Author Response

Please refer to the revision notes attached.

Reviewer 2 Report

Authors, congratulations on your work. I hope my recommendations help you improve this work even further. 

I reviewed the entire paper and suggest the authors make the update according to my comments.

 1.   I suggest briefly presenting a flowchart in the introduction, which shows the work stages. A steps flowchart enriches and also facilitates the reader to understand the method developed.

2.  In the introduction, describe the originality of the work What is the novel about this maintenance data management approach developed concerning other industrial data automatic management methods (it is not clear) to get better industrial management.?  What are the benefits for the industry? How this method assists maintenance managers to automate maintenance processes (industry 4.0 premise for maintenance management)?

3. Emphasize in the conclusion these points: (i) What is the novelty of this method for exchanging maintenance data using  AAS coupled with ISO 15926 RDL developed?  What is the economic industrial process gain (profits)? What are the maintenance management process benefits with the application of this exchange data (AAS with ISO 15926) method?

Author Response

(The authors gave the same response as above.)

Reviewer 3 Report

Interesting approach to show how Asset Administration Shell was improving the exchange of maintenance data. However, the conclusions of the manuscript can be improved. 

Author Response

(The authors gave the same response as above.)

Round 2

Reviewer 1 Report

In this work, the authors propose to study the use of an asset administration shell coupled with ISO 15926 to facilitate the exchange of equipment conditions and health status data of a process plant.

Through the proposed method, it is possible to transfer equipment data related to maintenance to a corresponding maintenance system in a neutral way and exchange the maintenance data between the maintenance systems regardless of equipment differences by manufacturers, replacement of equipment, and upgrade or change of maintenance systems. The research topic is relevant and definitely worthy of investigation. I believe, however, that the method ends up losing some of its potentials by limiting itself to studying only equipment conditions and health status data. I believe that, if it would consider other data necessary for decision-making related to maintenance (such as equipment criticality, maintenance policy, failure history, maintenance activity history, reliability, maintainability, availability, etc.), the method would really be more interesting from a practical point of view. As the authors say that such a task will be implemented in future works, I think that the present manuscript can be published as is.